# Applications of Proteomics in Ovarian Cancer: Dawn of a New Era

**DOI:** 10.3390/proteomes10020016

**Published:** 2022-05-09

**Authors:** Aruni Ghose, Sri Vidya Niharika Gullapalli, Naila Chohan, Anita Bolina, Michele Moschetta, Elie Rassy, Stergios Boussios

**Affiliations:** 1Department of Medical Oncology, Barts Cancer Centre, St. Bartholomew’s Hospital, Barts Health NHS Trust, London EC1A 7BE, UK; aruni.ghose@nhs.net (A.G.); naila.chohan@nhs.net (N.C.); 2Department of Medical Oncology, Mount Vernon Cancer Centre, East and North Hertfordshire NHS Trust, Northwood HA6 2RN, UK; 3Department of Medical Oncology, Medway NHS Foundation Trust, Windmill Road, Gillingham ME7 5NY, UK; 4Division of Research, Academics and Cancer Control, Saroj Gupta Cancer Centre and Research Institute, Kolkata 700063, India; 5School of Biosciences Education, Faculty of Life Sciences and Medicine, King’s College London, London WC2R 2LS, UK; sri.gullapalli@kcl.ac.uk; 6Department of Haematology, Clatterbridge Cancer Centre Liverpool, The Clatterbridge Cancer Centre NHS Foundation Trust, Liverpool L7 8YA, UK; anita.bolina@nhs.net; 7Novartis Institutes for BioMedical Research, 4033 Basel, Switzerland; michelemoschetta1@gmail.com; 8Department of Medical Oncology, Gustave Roussy Institut, 94805 Villejuif, France; elie.rassy@hotmail.com; 9School of Cancer & Pharmaceutical Sciences, Faculty of Life Sciences & Medicine, King’s College London, London WC2R 2LS, UK; 10AELIA Organization, 9th Km Thessaloniki-Thermi, 57001 Thessaloniki, Greece

**Keywords:** ovarian cancer, signaling pathways, proteomic biomarkers, proteomic techniques, multiomics, peptidomics

## Abstract

The ability to identify ovarian cancer (OC) at its earliest stages remains a challenge. The patients present an advanced stage at diagnosis. This heterogeneous disease has distinguishable etiology and molecular biology. Next-generation sequencing changed clinical diagnostic testing, allowing assessment of multiple genes, simultaneously, in a faster and cheaper manner than sequential single gene analysis. Technologies of proteomics, such as mass spectrometry (MS) and protein array analysis, have advanced the dissection of the underlying molecular signaling events and the proteomic characterization of OC. Proteomics analysis of OC, as well as their adaptive responses to therapy, can uncover new therapeutic choices, which can reduce the emergence of drug resistance and potentially improve patient outcomes. There is an urgent need to better understand how the genomic and epigenomic heterogeneity intrinsic to OC is reflected at the protein level, and how this information could potentially lead to prolonged survival.

## 1. Introduction

Ovarian Cancer (OC) in the United Kingdom (UK) is the sixth-most-common cancer as well as the sixth-most-common cause of cancer death among women [1]. Ninety percent of OC are of an epithelial cell type and comprise multiple histologic types, with various specific molecular changes, clinical behaviours, and treatment outcomes [2]. The remaining 10% are non-epithelial OC, which include, mainly, germ cell tumours, sex cord-stromal tumours, and some extremely rare tumours [3,4]. Finally, ovarian carcinosarcomas, accounting for only 1–4% of all OC, are composed of an epithelial as well as a sarcomatous component [5]. Serous papillary peritoneal cancer is a clinically well-recognized but biologically enigmatic entity. It shares common molecular, histological, and clinical features with epithelial OC, mainly high-grade serous OC, which has led to similar management of the two entities [6]. OC is the most lethal gynaecologic malignancy among women of advanced age (>40 years), especially in developed countries [7]. Fifty-eight percent of patients are diagnosed at an advanced stage and are associated with a five-year survival of 27% for stage III and 13% for stage IV disease [1,7]. The asymptomatic growth of the cancer and the lack of an effective screening endeavour renders it a “silent killer”.

To date, most OC screening programmes have used serum biomarker cancer antigen 125 (CA125) and transvaginal ultrasound scan (TVS) [7]. When used individually, as in the Prostate, Lung, Colon and Ovary (PLCO) Cancer Screening Trial, neither modality was optimally sensitive or specific [8]. Following this, two-stage sequential strategies developed during the Normal Risk Ovarian Screening Study (NROSS) and the UK Collaborative Trial of OC Screening (UKCTOCS) [9,10]. On elevation of CA125, leading to an increased Risk of OC Algorithm (ROCA) score (first stage), TVS would be indicated (second stage), which if abnormal, would warrant a surgical intervention. Both the above trials demonstrated superior specificity and sensitivity for screening [9,10]. The UKCTOCS yielded a 20% reduction in mortality in patients with an average risk [10].

Germline mutations of the genes *BRCA1/2* are related to increased cancer predisposition, and they account for approximately 14% of epithelial OC [11]. Moreover, cancers arising from the fallopian tubes (about 70%) and high-grade serous cancers are usually associated with *BRCA1/2* mutations [12]. This predisposes to a late diagnosis, and, therefore, early detection in stage I or II becomes a significant challenge. Moreover, in the initial stage of screening, around 20% of OC diagnoses are missed using CA125 alone [13].

This led to the unmet need for identifying biomarkers, to increase the sensitivity of the first stage. The Human Epididymis Protein 4 (HE4) is a protein antigen (PA), noted to have better specificity than CA125 in detecting early-stage OC, as it could easily differentiate malignant from benign pelvic masses [14]. Hence, their combination was used in the Risk of Malignancy Algorithm (ROMA). The OVERA test constituted the above, as well as other Pas, including transferrin, apolipoprotein A1, and follicle-stimulating hormone. Both ROMA and OVERA improved first stage sensitivity, while maintaining comparable specificity to ROCA [15]. They were supported by the Early Detection Research Network (EDRN) and gained United States Food and Drug Administration (FDA) approval in 2011. Other than PAs, the search for new biomarkers is an ongoing process, including autoantibodies (anti-TP53), microRNA (miRNA), circulating tumour DNA (ctDNA), and methylated ctDNA.

The process of development of a new biomarker proceeds through three stages namely discovery; assay development and analytical validation; and clinical validation and utility [16]. In the biomarker discovery phase, traditional methods are being replaced by omics techniques such as genomics, epigenomics, transcriptomics, metabolomics, and proteomics. However, only a select few biomarkers have been FDA approved, especially in OC, due to the paucity of validation tools from their discovery in the lab to implementation in the clinical setting [17]. In our review article, we provide an overview on how targeted qualitative and quantitative proteomic technologies have impacted biomarker development for early detection of OC.

## 2. Signaling Pathways in OC

Several signaling pathways influence multiple cellular processes in epithelial OC and, especially, the pathogenesis, as it is demonstrated in Figure 1. Thorough understanding of the precise role of these pathways can lead to the development of new and more effective targeted therapies as well as novel biomarkers in OC.

A summary of the number of relevant studies on ovarian cancer biomarkers, according to their phase, is illustrated in Figure 2.

Lysophosphatidic acid (LPA) is implicated in OC pathogenesis, including tumour progression, migration and invasion [18]. It is, also, implicated in ascites formation and tumour angiogenesis. The action of LPA is thought to be mediated through LPA receptors, and high levels of LPA are expressed in OC, thus making it a potential therapeutic target [19].

Phosphatidylinositol 3-kinases (PI3K)/AKT/mTOR is a vital signalling pathway involved in the regulation of cellular processes such as growth, metabolism, and survival. Hyperactivation of this pathway is associated with OC, particularly that of endometriod and clear cell carcinomas [20]. Several studies have shown hyperactivation and dysregulation of this pathway contribute to OC cell proliferation, migration, and chemoresistance, especially through the somatic mutations or amplifications of phosphatidylinositol-4,5-bisphosphate 3-kinase catalytic subunit alpha (*PIK3CA*) [21].

The activation of nuclear factor kappa-light chain (NF-kB) pathway in OC promotes aggressive tumour behaviour such as invasion, metastasis, and chemoresistance. NF-kB is comprised of a group of transcription factors; the composition of such transcription factors drives the pathway down the canonical, with ultimate activation of either p65/p50 or noncanonical with activation of Relb-p52 [22]. The canonical pathway is implicated into the proliferation and angiogenesis of cancer, whereas the role of the non-canonical pathway is yet to be established, but is thought to act on cancer-stem-cell-like properties [23].

Mitogen-activated protein kinase (MAPK) pathways are involved in cellular processes including proliferation, differentiation, and apoptosis of cells in response to external stimuli [24]. Studies have shown that continuous activation of this pathway can transform normal cells into tumour cells. Activation of MAPK in OC, frequently reported, is associated with worse clinical outcomes [25].

Sarcoma proto-oncogene tyrosine-protein kinase Src is a downstream component of growth factor receptors and is overexpressed in OC [26]. It is thought to promote chemoresistance, and studies have shown the inhibition of Src results in enhanced apoptosis by chemotherapy agents such as paclitaxel and docetaxel, thus making it a biological target of interest [27].

The epidermal growth factor receptor (EGFR) family—homologous with human epidermal growth factor receptor (HER) or ERBB—when activated, ultimately, leads to cellular responses such as proliferation, differentiation, and cellular survival, via propagation of intracellular cascades, including MAPKs and AKT [28]. Activation and increased expression of such pathways are associated with tumour angiogenesis, differentiation, metastasis, and resistance to apoptosis [29].

Vascular endothelial growth factor (VEGF) is vital for angiogenesis, and its overexpression is implicated in OC tumour progression, by promoting tumour angiogenesis and vascular permeability [30]. Studies have shown high VEGF levels to be correlated with poorer outcomes [31]. Interest in antiangiogenic therapies and their significance remains ongoing.

Signal transducer and activator of transcription 3 (Jak-stat 3) remains a promising therapeutic target for OC, as it is frequently overexpressed. Jak-stat 3 plays a role in tumour angiogenesis, survival, invasion, and chemoresistance [32]. Inhibition of this pathway is shown to suppress tumour growth and progression [33].

Interleukin-6 (IL-6) is a key proinflammatory cytokine associated with promoting invasion, tumour survival, chemoresistance, and angiogenesis via VEGF overexpression in OC cells [34]. Further to this, IL-6 is implicated in the activation of JAK-STAT 3 pathway, which is associated with tumour growth and chemoresistance [35].

## 3. Proteomic Biomarkers for OC

### 3.1. Tissue Proteomics

Cancer has proteomic heterogeneities, wherein a genetic alteration in normal cells disrupts the functional networks of encoded proteins that power cell survival, growth, invasion, and metastasis [36,37,38]. Since it is a heterogeneous disease, it involves several tumours with different histopathological features. OC has over 30 different types, each of which originates from a different cell and has its own distinct proteome [39,40,41].

For this reason, the diagnosis and prognosis of the disease are beyond the scope of microscopic analysis. Due to this variety in tumours, the treatment should be customized, giving each patient a specific inhibitor that disrupts various points of the pathway (patient-tailored combination therapy), which would function, in theory, while decreasing toxicity and increasing efficacy. However, to do this there must be a direct correlation with the in vivo scenario, as the analysis of cultured cell lines without enrichment will be insufficient [38].

Fortunately, technology in tissue proteomics, though with some shortcomings, currently, provides a solution. Laser capture microdissection (LCM) allows selection of specific subpopulations from a single biopsy specimen such as a tumour, premalignant cells, or normal cells, for proteomic analysis of the tissue microenvironment and, thereby, aiding molecular profiling. Following this, various analytical methods can be executed, such as immunoprecipitation, gel electrophoresis, immunoblotting, and histochemistry [42].

However, the most popular technology now being used is mass spectrometry (MS), in league with protein microarrays. The reason these go hand in hand is due to their complementarity. MS requires no prior knowledge regarding the identity of the protein to be investigated. To an extent, it provides knowledge on both defects in the post-translational protein, and on its identity. In contrast, some part of the protein’s identity must be known for protein microarrays, however, they can be run with a small input material, unlike MS, which cannot analyse clinical biopsy in lieu of its size [38].

Several challenges come to play in tissue proteomics, the most important of all being the invasive technique.

### 3.2. Proteomics of Post-Translational Modifications

Proteomic analysis is a key player in the detailed comprehension of biological systems. It does not, however, take into consideration the high dynamic range of samples and expression of profiling based on miRNAs and the full complexity of the proteome [43,44]. Recent studies have shown the potential of post-translational modification (PTM) analysis, such as glycosylation, phosphorylation, acetylation, and methylation, as a powerful strategy for the discovery of biomarkers and the understanding of regulatory mechanisms, cell signalling, communication, and adhesion [45,46].

Protein Glycosylation is a complex and common PTM, responsible for several vital processes, such as protein localisation, folding, trafficking, solubility, antigenicity, half-life, cell communication, signalling, and adhesion [47]. It comprises four main categories based on the glycan base, namely O-linked glycosylation, N-linked glycosylation, glycol-phosphatidylinositol anchor attachments, and C-mannosylation [48]. Characterisation and analysis must be done on the two main groups, N-linked glycans and O-linked glycans, after targeted isolation, enrichment methods, and sensitive detection, due to the challenges posed by the glycans variety and structural complexity [45].

Some techniques proposed for glycoprotein enrichment are chemical modifications such as immunoprecipitation or affinity chromatography, done by entrapping peptides and proteins with negatively charged phosphate groups onto a positively charged matrix. Examples include lectin chromatography, hydrophilic interaction liquid chromatography, hydrazide chemistry, or capture through immobilized titanium dioxide and boronic acid [44].

The characterisation of the proteome in patients with OC has been improved using glycomics [44,45]. Abbott et al. studied the differences in the transcripts of a small set of glycosyltransferases involved in the N-linked glycosylation pathway in normal ovarian and cancerous tissue [49]. This was done by verifying glycoproteins in patient sera through immunoprecipitation and microarray methods. Results showed periostin and thrombospondin to be potential biomarkers for OC. Shetty et al. probed into N-linked sialylated glycoproteins in OC and showed that the upregulation of 10 N-linked glycopeptides, PON1, haptoglobin, in the patient’s serum was indicative of OC [50]. Soon after, sialic acid containing glycoproteins (sialome) were identified as novel biomarkers in ascites and ovarian cyst fluid [51]. Saldova et al. showed that the differentiation between CA125 glycoforms and controls improves the specificity and sensitivity of CA125 [52]. Recently, a novel method coupling microarrays and HILIC-UPLC helped reveal a monoclonal A4 antibody structure, which, like other cancer-specific antibody bindings to glycan studies, helps simplify use to indicate the diagnosis [53].

The next category, reversible protein phosphorylation, the highly dynamic, mainstay of intracellular signalling networks and phosphoproteomics, is the optimal choice for studying these networks, i.e., proliferation, apoptosis, homeostasis, or metabolism [54]. Phosphoproteins such as glycoproteins are normally of low concentration in biological samples, due to which similar enrichment techniques (affinity chromatography and immunoprecipitation) must ensue to enable phosphorylation. Another reason for enrichment is due to the tendency of phosphoproteins to co-exist with their unphosphorylated isoform in the cell. Phosphorylation provides a promising strategy for the understanding of molecular determinants of OC. Francavilla et al. used this technique to show that cell proliferation is controlled by cyclin-dependent kinase 7 (CDK7), through an examination of epithelial cells from OC and healthy patients [55].

As a result, it helped conclude that inhibiting CDK7 may aid in the development of effective therapeutic techniques. This has also unearthed several signalling pathways involved in OC pathogenesis, such as the activator of transcription 3 (Jak-STAT 3), proto-oncogene tyrosine-protein kinase Src pathway, NF-kB, MAPK, ErbB activation, PI3K, lysophosphatidic acid, EGF and VEGF, Mullerian inhibitory substance receptor, and ER-beta pathways [56].

This pathway-based approach can create large improvements in the treatment and diagnosis of cancer, through further proteomic studies that are necessary to validate the information above [46]. Although analysis of post-translational modifications appears to be a reliable and successful method for identifying biomarkers and studying cell signalling networks, there are certain drawbacks to be aware of. Firstly, the changes must be released chemically or enzymatically. Secondly, reliable study results almost always necessitate derivatization. Thirdly, sample preparation might be challenging because of the variability and small concentration of post-translational modifications. Finally, MS approaches have a restricted dynamic range, which means that tiny changes are not detectable, and, even when correct detection is obtained, precise spectra interpretation and protein structure assembly are difficult to achieve [43].

### 3.3. Quantitative Proteomics

Over the last decade, proteomics based on mass spectrometry (MS) has emerged as a promising tool for revealing the quantitative condition of the human proteome [57]. The introduction of quantitative techniques has opened new avenues for investigating the differential expression of a protein as well as posttranslational and posttranscriptional alterations in various situations, bettering the understanding of the functional ramifications of changing gene expressions [46]. Quantitative proteomics based on MS can be divided into 2 categories: bottom-up, i.e., measuring peptides as surrogates of the protein of interest, and top-down, i.e., measuring the whole protein. The bottom-up approach is mostly used for biomarker development [57].

In the discovery stage of the biomarker development workflow, many biomarker candidates are identified from a few sample groups, by using the first type of quantitative approach, untargeted quantitative proteomics [57]. During the next step, the verification stage, a small number of biomarker candidates are further evaluated for reproducibility in many sample sets. Finally, the biomarker candidates, which are the most promising, are validated in a much larger number of sample cohorts, to assess their sensitivity, specificity, and clinical utility, which is done by using the second quantitative approach, targeted quantitative proteomics [57].

Label-free and stable isotope labelling techniques using a data dependent acquisition (DDA) mode are examples of untargeted quantitative proteomics approaches that are intended to yield an in-depth unbiased quantitation of the global proteome in the discovery stage [58]. Stable isotope labelling quantifies peptides at the precursor ion (MS1) level, or peptide fragments at the production ion (MS2) scan level, by utilising the mass increase caused by the mass tags with incorporated stable isotopes [57]. Chemical labelling strategies, such as Isobaric Tags for Relative and Absolute Quantitation (iTRAQ) and Tandem Mass Tags (TMT), and metabolic labelling strategies such as Stable Isotope Labelling by Amino Acids in Cell Culture (SILAC), are two methods for the labelling of peptides or proteins with stable isotopes [57].

SILAC is a popular method for the analysis of OC lines [44]. It was used to show the influence of the urokinase plasminogen activator on OC cells, the suggested therapeutic use of calcium-activated chloride channel regulator 1 and chloride channels in OC, and so forth [44].

Data independent acquisition (DIA) methods are also useful in quantitative studies. It is a platform-dependent technology and is a targeted data extraction quantitative. It is suitable for the discovery of biomarkers and has some advantages in terms of ease in assay development and its analyte multiplexing ability. Some examples of the technology are SWATH, MSE, diaPASEF, SONAR, BoxCar, etc. DIA is also often used, along with DDA, in label-free quantification [44,57,59].

Label-free quantification, for peptide and protein quantification, uses mass spectrometric signal intensity or peptide spectral counts. DIA, in this case, uses MS1 first and acquires MS2 scans from all identifiable peptides from the detection window of MS1. Whereas, when utilising DDA, a set number of precursor ions from the most abundant peptides from the MS1 complete scan are chosen to obtain MS2 scans [57,59].

Global proteome analysis using untargeted quantitative proteomics yields promising relative quantitation data for many biomarker candidates in biomarker discovery. However, due to the stochastic nature of abundance-based precursor ion selection in the DDA mode, there is no guarantee that the same peptides will be consistently detected in all analyses. This strategy, in addition to low reproducibility, also has higher missing values, if data imputation or DIA are not used, for low abundance peptides/proteins [57]. These limitations confirm that untargeted global quantitative proteomic approaches are unsuitable for biomarker candidate verification and validation. Therefore, targeted quantitative proteomics methods are used to verify and validate biomarkers.

### 3.4. Biofluid Proteomics

A biomarker for early detection of OC should ideally be detected in a biological sample that is easily accessible to a physician and should not require large invasive procedures [60]. Body fluids such as plasma, serum, ascites fluid, pleural effusions, ovarian cysts, and urine meet these criteria and have, thus, been recommended as ideal sources for biomarkers.

They are a good source for evaluating a biomarker before any additional clinical symptoms are detected. However, proteomics-based biomarker discovery is complicated due to the biological makeup of plasma, urine, and other bodily fluids. The plasma proteome, for example, contains proteins with a wide range of concentrations (nine orders of magnitude), with a few high-abundance proteins such as albumin, immunoglobulins, transferrin, 1-antitrypsin, and haptoglobin accounting for around 95% of the total protein content in plasma [60].

The first serum biomarker, identified in 1965, was carcinoembryonic antigen (CEA), which was used to detect mucinous tumours [61]. Cancer antigen 125 (CA125), a glycoprotein released naturally by epithelia of several organs, found in 1981, is the most used and studied biomarker in endometrial and serous OC. Changes in it can, also, correlate with the progression of the disease and can be assessed through monitoring serial CA125 eligibility for secondary cytoreductive surgeries [46,61]. However, despite its popularity, it only has a positive predictive value of 4% and is neither sensitive nor specific enough to use alone to detect serous epithelial cancer arising from the ovary, fallopian tube, or peritoneal cavity.

For that reason, more biofluid biomarkers have been assessed, such as the Human epididymis protein 4 (HE4), mesothelin, osteoponin, proatasin, lysophosphatidic acid, macrophage colony-stimulating factor, etc. [46]. HE4, discovered in 2008, is currently used primarily to monitor the recurrence or progression of epithelial OC and is found to be overexpressed in endometroid (100%), serous (93%), and clear cell (50%) tumours. In comparison to CA125, HE4 has higher specificity in premenopausal cases and benign conditions, as well as higher sensitivity in early-stage tumours. Furthermore, it is overexpressed in 32% of cases with non-elevated CA125. However, it cannot detect mucinous tumours, and there are discrepancies in the results of combining CA125 and HE4, based on a review of the literature [46,61]. However, the FDA approved the Risk of Ovarian Malignancy Algorithm (ROMA), which is a combination of serum CA125, HE4, and menopausal status, to distinguish benign masses from malignant tumours.

Early biomarker studies greatly relied on the surface-enhanced lased desorption MS (SELDI-MS), which has low reproducibility and does not recognise the copious peptide species. Later studies have helped in the identification of more biomarkers that can be used for the early detection of OC, such as In vitro Multivariate Index Assay (IVDMIA), which uses several markers together to improve performance in a clinical setting [46,61].

OVA1 was the first IVDMIA that was approved for use in 2009 by the FDA. It consists of five serum proteins, namely transthyretin, β-2 microglobulin, CA125, transferrin, and apolipoprotein A1. In both pre-menopausal and post-menopausal women, OVA1 detects 94% of cancer cases, while CA125 detects 77%. It is more sensitive than CA125 alone, but has a lower specificity (54%) and should not be used to predict the risk of OC in asymptomatic patients without pelvic masses. The purpose of this test is to determine the likelihood of malignancy in women who present with an ovarian adnexal mass before planned surgery.

Another shortcoming of this test is its higher false-positive outcome. Therefore, OVA1 was upgraded to OVERA (approved in 2016 by the FDA), where transthyretin was replaced by HE4 and β-2 microglobulin by follicle-stimulating hormone (FSH). These protein biomarkers were discovered using immunoassay-based techniques, such as enzyme-linked immunoassay (ELISA) and radioimmunoassay (RIA) [61]. OVERA is designed to differentiate between patients at risk of malignant and those with benign tumours. It also maintains high sensitivity (91%) while outperforming OVA1, in terms of specificity (69 vs. 54%) and positive predictive value (40 vs. 31%) [61].

Table 1 depicts the biomarker discovery from fluid-serum/plasma, using different platforms proteomics in separate population.

Table 2 summarises ovarian cancer biomarkers detected by proteomic techniques as part of prospective cohort studies, specifying the number of patients and controls, along with sensitivity and specificity.

Finally, Table 3 depicts mechanistic biomarkers in ovarian cancer, with reference to patients’ sample-based studies.

## 4. Proteomic Techniques

All the various types of proteomics and biomarkers described in the previous section come down to the method/technology used to extract and identify proteomes, cells, or other substances. These techniques, also, caused the limitations seen most often. So, finding an ideal proteomic analysis procedure with a low margin of error is of prime significance.

### 4.1. Two-Dimensional Gel Electrophoresis (2DE)

Proteomic analyses were, first, carried out by the traditional two-dimensional gel electrophoresis (2DE), which separates sequentially, mid-range molecular weight proteins based on two distinct characteristics of the protein: size and charge [43]. It has been mentioned that proteome analysis with 2DE techniques was later expanded with liquid chromatography-mass spectrometry (LC-MS) techniques. Despite the rapid advancement of MS-based proteomics, 2DE continues to play an important role in the areas of protein identification from organisms with no or incomplete genome sequences, alternative detection methods for specific proteomics modification, and identification of protein isoforms and modified proteins. This proves that there is a significant market for 2D gel-based proteomics, which is supplemented by traditional LC-MS techniques [43]. The protein contents of tissue or biofluids are separated using two-dimensional gels and must be visualised using staining. Silver staining used to be employed for this purpose, but, as of late, other dyes such as Deep Purple and Sypro Ruby, or newer staining techniques, namely fluorescent labelling (CyDye), are being utilised [91]. For protein labelling, the CyDye method employs three different fluorescent dyes with different emission spectra: Cy2, Cy3, and Cy5. Three differentially labelled samples are run on a single gel, and a gel image is generated for each of the three dyes, due to their distinct emission spectra. These images are, then, superimposed on top of one another, and differences in protein levels are analysed using sophisticated software such as Prognosis (Nonlinear Dynamics, Newcastle, UK) or DeCyder (GE Health, Little Chalfont, UK) [91]. One significant advantage of this method is that the three differentially labelled samples produce perfectly matching gel images, removing gel-to-gel variability. The introduction of CyDye vastly improved reproducibility and detection sensitivity in the picogram range, due to the ability to run disease and non-disease samples, as well as internal control, on a single gel [91]. The proteins can, later, be identified using MS, after differences in protein levels of candidate markers between disease and control groups have been determined [91]. The best ‘snapshot’ of the protein repertoire of the cell or body fluid has come from 2DE separation. However, this technology, on its own, is limited because it has low throughput, is time-consuming, is labour intensive, and has difficulty detecting proteins with a basic charge or those smaller than 10 kDa [92].

### 4.2. MS-Based Techniques

In proteomics, MS is the most-used protein identification technique. It is utilised for fingerprinting proteins and peptides, after which direct sequencing can be done [93,94]. It helps in precisely determining the mass and charge (*m/z*) of proteins and, as a result, identifying the exact progenitor proteins. MS instruments have substantially improved in recent years and are now widely used to detect even small samples, as they are sensitive to the picomole- femtomole range, necessary for oligonucleotide, glycoprotein, and other minuscule molecule detection [43].

An ion source, a mass analyser, and a detector make up a mass spectrometer. Ionization sources, such as matrix-assisted laser desorption and ionization (MALDI) and ESI; mass analysers, such as time-of-flight (TOF), quadrupole, and ion traps; and fragmentation methods, such as collision-induced dissociation (CID), electron-transfer dissociation (ETD), and electron-capture dissociation (ECD), can all be used in various combinations in MS [43]. The most widely used separation method for studying biological samples by MS or MS/MS (tandem mass chromatography) is high-performance liquid chromatography (HPLC) [43].

The shotgun-MS method entails the separation of peptides by protein digestion in complex mixtures, tandem mass spectrometric analysis, and, finally, running the process through databases to identify the peptides. The initial step of MS is to lyse the sample and extract the proteins from the sample. This is followed by the digestion of the proteins into peptides, resulting in the yield of thousands of peptides, which are then enriched in various ways (for example, affinity resins or specific antibodies) or pre-fractionated, based on their physicochemical properties (such as charge or isoelectric point). These samples are, later, individually examined using reversed-phase liquid chromatography added to MS (LC-MS). Next, MS/MS or MS is connected to software, which analyses the identified ions based on their relative abundance and *m/z*. Finally, these ions are uploaded/cross-verified in the databases, enabling the identification of the fitting peptides sequences [43].

The dynamic range, rather than sensitivity, is the most difficult challenge in most MS approaches. To increase the amount of information that can be obtained from specific samples, the common proteins or peptides can be removed. Intriguingly, common proteins such as albumin can also act as carrier proteins, capturing a subset of proteins and peptides helping achieve optimal results [93]. Recently, the use of MALDI and SELDI MS techniques have skyrocketed, due to their higher rates of accuracy.

#### 4.2.1. MALDI-TOF

MALDI with TOF is a technique that uses laser excitation to present proteins in a matrix. In the MALDI-TOF detection, samples of interesting protein are immobilised on an energy-absorbing chemical matrix on a chip or plate. This method is followed by the analysis of the entire proteome, and presentation of the proteins coupled in a matrix to laser excitation. The peptides, which are ionised by this process, are, later, sent to the detector through a vacuum tube. Like the traditional MS, peptides are detected in order of their *m/z*. Based on the time each peptide reaches the detector, a peptide fingerprint is formed that reflects the sample’s protein composition and the relative abundance of specific proteins. In many circumstances, guided proteolysis is employed to reduce the size of the peptide, to put the charge to mass ratio into the ideal range for the MS system in question. Peptide mass fingerprints are compared to huge, published databases and masses predicted by protein sequences to identify proteins or peptides [93,94].

Another MALDI technique is MALDI-MS imaging (MALDI-MSI). Here, mass spectrometry imaging (MSI) combines immunohistochemistry, fluorescence microscopy, and MALDI-MS instrumentation, to provide a specific molecular image of numerous expressed proteins within a tissue sample. To perform MALDI-MSI, sections of biological tissues are introduced into a MALDI-MS instrument, and the ultraviolet-pulsed laser of the MALDI source is used to raster over a selected area, while acquiring mass spectra of the ablated ions at each image point [93,94]. However, MALDI-MSI has not yet been used to map the transcriptome, including miRNA and other RNA-related molecules [43].

There is now a broad field of clinical research to investigate this, known as tag-based mass spectrometric imaging (Tag-MSI) or targeted mass spectrometric imaging (Tag-Mass MSI), which supplements the MALDI-MSI [43].

#### 4.2.2. SELDI-TOF

SELDI is a refinement of MALDI that uses a selective surface to bind a subset of these proteins based on absorption, partition, electrostatic interaction, and/or affinity chromatography. Artificial intelligence techniques are needed to sort or mine the data and extract critical information because a SELDI proteomic profile can now have up to a million data points (high-resolution spectra of 400 ppm) [94,95].

SELDI-TOF technique employs stainless-steel- or aluminium-based solid supports, with predetermined bait sections comprising a variety of surface-binding chemistries, such as hydrophobic, normal-phase, metal-affinity, cationic, or anionic, as bait in a 1–2 mm diameter area. It does not require prior purification or protein fractionation, and cell lysates or bodily fluids as tiny as 0.5 μL can be applied to these surfaces [92]. Proteins and peptides are, selectively, kept on the protein chip after a wash phase; based on the bait used and the unique chemistry of each protein, they are then evaluated using mass spectrometry TOF technology. Similar to MALDI, ionized proteins and peptides are recorded when they impact the detection plate, and the variation in the journey time down the vacuum tube is recorded [92]. It can profile proteins, regardless of their intrinsic hydrophobicity (a limitation of 2DE analysis), and is extremely sensitive in detecting proteins in the lower-molecular-weight range. Finally, data-mining methods examine low molecular weight (0–20 kDa) proteomic data streams generated by SELDI-TOF, sorting the data into homogenous (control or illness) groups [92].

### 4.3. Protein Microarrays

Protein microarrays investigate how proteins are expressed and activated, thereby providing a functional view of protein networks [94]. Microarray technology has allowed researchers to assess the expression of tens of thousands of genes in a single tissue sample and compare normal and abnormal gene expression, such as the difference in the regulation, during cancer formation, in malignant cells using antigen-antibody interactions.

Protein microarrays can be divided into two major categories: forward phase arrays (FPAs) based on cell lysate probing and reverse phase arrays (RPAs) based on antibody probing [43].

In both forms of microarrays, a protein substrate is immobilised and queried using a tagged probe. Protein expression is, then, quantified based on the strength of the resultant signal. However, in RPAs, antibodies are immobilised, as bait molecules, to catch proteins from the biological milieu. FPAs, on the other hand, necessitate the immobilisation of cellular lysates, which are, subsequently, probed with particular antibodies for the proteins of interest [94].

RPAs, unlike FPAs, do not necessitate the labelling of cellular protein lysates and, thus, provide a useful platform for biomarker screening, therapeutic monitoring, and pathophysiologic studies. Furthermore, the RPA is unique in its ability to analyse signalling pathways, utilising small numbers of cultured cells or cells isolated via laser capture microdissection (LCM) from human tissues obtained during clinical trials [43].

The RPA platform has helped in identifying therapeutic targets, the study of disease progression, and profile signalling pathways, as well as suggesting prognostic indicators in OC. A major drawback of RPAs is that they necessitate the use of particular antibodies, so there remains a perpetual risk of non-optimal antigen-antibody interaction, which could lead to incorrect results. Therefore, these approaches should be considered purely exploratory [43,94].

### 4.4. Mitochondrial Proteomics Methods

In eukaryotic cells, the mitochondrion is a highly specialised organelle with vital biological activities. It is a multifunctional organelle that is responsible for energy metabolism, oxidative stress, cell apoptosis, cell cycles, mitophagy, and communication with other organelles. Over the last few decades, research has led to the identification of mitochondria-related signalling pathways for insights into the mitochondrial mechanisms in tumourigenesis, identifying biomarkers for diagnosis and prognostic assessment, and discovering therapeutic targets for effective therapy [96].

Since the first human placental mitochondrial proteome was analysed with MS in 1998, great advances in mitochondrial proteomics have been made [97]. The reason for the probing into the mitochondrion is not only due to its multifunctional characteristics, but also because it is the only organelle other than the nucleus to contain independent DNA: mitochondrial DNA (mtDNA). There is compelling evidence that mtDNA influences tumour progression cell functions and susceptibility in the tumour microenvironment, and several studies have found that mitochondrial dysfunction is closely linked to OC [98].

A study by Hecker et al. of mitochondria in cells with OC under an electron microscope showed that the loosened mitochondria have a moth-eaten appearance, which indicated their involvement in OC [99]. Another study by Dier et al. reiterated this fact, by proving that altered mitochondrial fission dynamics in the mitochondria affected the phenotype of specific epithelial OC cells [100]. Some mitochondrial proteins, such as Bcl2, p53, and galectin3, have also been reported to be therapeutic targets used to suppress mitochondria-related pathways in OC [101,102,103]. As a result, quantifying mitochondrial proteome changes will shed light on any mitochondria-mediated pathophysiological changes in OC.

The importance of high-purity mitochondrial samples in proteomic analysis cannot be overstated. Many methods can be used for this purpose, including kit-based methods, free-flow electrophoresis, kit-based methods, and density-gradient centrifugation. Multidimensional protein identification technology (MudPIT) combines strong cation exchange (SCX) prefractionation, reversed-phase high-performance LC separation, and MS to increase the number of identified peptides and the power of peptide separation [104]. After which, to isolate the mitochondria, quantify mitochondrial proteins, and quantify mitochondrial phosphoproteins, several quantitative proteomics methods can be utilised, such as the SILAC, ICAT, iTRAQ, tandem mass tags (TMT), and label-free methods.

Using these techniques, the mitochondrial proteome in OC has a quantitative reference map established, especially, for those that are platinum-sensitive (A2780) and platinum-resistant (A2780-CP20) [105]. In addition, around 5115 mitochondrial expressed proteins (mtEP) have been identified [106]. Further analysis of the mtEPs has revealed several potential biomarkers and signalling pathways in OC, such as CPT2, PKM2 (overexpressed in cancerous tissues), and HMGCS2.

The protein phosphorylation profile of mitochondria has also been identified, to understand the molecular mechanisms in pathophysiological conditions, thanks to significant advances in MS-based proteomics [107]. For example, phosphorylated cofilin 1 (p-CFL1) was found to be highly expressed in paclitaxel-resistant OC cells and chemoresistant cases relative to chemosensitive ones from primary human OC tissue. As a result, it can be stated that mitochondrial proteomics in conjunction with clinical data are a valuable resource to develop a prognostic model in OC.

## 5. New Approaches in Proteomics

### 5.1. Targeted Proteomics

Targeted proteomics is a key technique that enables the validation and verification of biomarkers that have been discovered. It works with untargeted proteomics to complete the cycle of biomarker discovery and validation. It consists of two types, –multiple reaction monitoring (MRM) and parallel reaction monitoring (PRM), which can facilitate quantitation accuracy during the validation and verification process of biomarker discovery [108].

MRM is a traditional targeted quantitative proteomics approach that employs either triple quadrupole (QqQ) or quadrupole linear ion trap (QTRAP) instruments [109]. The targeted distinctive precursor ions for peptides of interest are selected in the first quadrupole (Q1) and, then, transmitted into the collision cell (Q2) for fragmentation. Finally, specific product ions derived from the targeted precursor ions are selected for detection in the third quadrupole (Q3). Q3 is, generally, a low-resolution mass analyser that cannot transmit ions with isolation widths greater than 0.7–1.0 Da, without losing sensitivity. MRM detects transitions (Q1/Q3 MRM ion pairs) by utilising the unique features of a QqQ or QTRAP instrument [110].

PRM is a newer targeted acquisition method and uses high-resolution accurate mass (HRAM) analysers, such as Orbitraps. In contrast to the QqQ-dependent MRM methods described above, Q3 is replaced with an HRAM mass analyser, enabling the parallel detection of all productions from targeted precursor ions, rather than selecting a limited number of productions (usually three transitions). PRM generates highly specific spectra for all productions derived from selected precursor ions, allowing for high-confidence targeted peptide identification. Since it uses several transitions to identify and quantify peptides, the predetermination of transitions and collision energy, which is required for MRM, is not required, reducing method development time. As opposed to MRM, PRM provides a significant improvement in signal-to-noise, while maintaining high sensitivity [111].

Targeted proteomics can be used to bridge the chasm between biomarker candidate discovery and clinical utility. They allow us to look at the complete proteome, or a sub-proteome at the same time, to see whether there are any links between protein expression (or alterations) and disease development. One of the most promising areas is the analysis of distinct protein patterns linked with ovarian cancer as a discovery technique for prospective biomarkers. The tumour–host communication system is directly influenced by the proteomic tissue microenvironment, making it a viable source for biomarkers. Given the time and money required to bring a medicine to market, the availability of biomarkers capable of detecting probable drug failures early in development is critical. Multiple biomarkers will be required for accurate screening and diagnosis. Understanding oncogenic signalling and generating biomarkers predictive of patient outcome in response to specific medication regimens is another aspect that has considerable promise for therapeutics. Proteomic study can help us understand and prevent adaptive response in cancer cells, such as epigenetic changes and protein network reorganization, through post-translational modifications. However, like all other proteomics, targeted quantitative proteomics has some limitations—the throughput is constricted at around 50–100 proteins per analysis, and it has a low sample multiplexing ability [57].

### 5.2. Peptidomics

Peptidomics is a new sub-division of proteomics and can, also, be used to shed light on new biomarkers. It is the study of peptides used to determine the exact form of each peptide. Like proteomics, it is used to identify new peptides that exist within tissue. It utilises quantification techniques to measure the relative level of peptides in varying environments.

Several studies have been done on peptides in oncology. Villanueva et al. communicated the differences between serum peptides in patients diagnosed with cancer and control experiments [112]. Lopez et al. discovered a panel of serum peptides that distinguished stage OC cancer from controls [113]. Fredolini et al. outlined 59 serum peptidome markers that are differentially expressed in OC versus benign conditions [114]. Bery et al. relayed the peptide markers that are expressed in the ascites fluid of patients with OC against controls [115].

Since proteolytic processes are deeply rooted in oncogenesis, peptidomics is expected to grow in the coming years with more advances in technology and understanding.

### 5.3. Exosomes

Exosomes, which are secreted by different cell types and are as small as 30–150 nm vesicles, play a critical role in intercellular communication. They have emerged as a compelling diagnostic and prognostic biomarkers for OC, as they may transport some tumour-associated proteins [46]. These organelles can be found in various body fluids, such as urine, blood, CSF, breast milk, and saliva, making them more ideal, due to their being less invasive.

Exosome research provides large-scale protein analysis when using MS-based proteomics. For example, Sinha et al. used an MS-based technique to characterise isolated exosome proteomes of OC cell lines [116]. However, a more recent study by Zhang et al. was the first-time exosomes were extracted from patient serum using LC-MS/MS and tandem mass tagging (TMT) [117]. This study used enriched exosomes from plasma and characterised them using technologies such as dynamic light scattering (DLS), nanoparticle tracking analysis (NTA), Western blot analysis, and transmission electron microscopy (TEM). The exosomal marker proteins CD81 and TSG101 were noted to be succinctly stained in the exosome samples, whereas the endoplasmic reticulum protein calnexin was not. Through this process, a net total of 294 exosomal proteins were identified. Two hundred and twenty-five of these proteins were found to be present in both the cancerous and non-cancerous samples. Exosomes isolated from the serum of women with various stages of OC were found to have higher levels of expression of 8 miRNAs (miRNA-21, miRNA-141, miRNA-200a, miRNA-200b, miRNA-200c, miRNA-203, miRNA-205, and miRNA-214) than those isolated from the serum of women with benign diseases, implying that miRNAs of circulating tumour exosomes can potentially be alternative biofluid biomarkers. Eventually, four genes (*LBP*, *FGG*, *FGA*, and *GSN*) were selected to be potential diagnostic and prognostic biomarkers because of their involvement in the apoptosis and coagulation pathways. The proteomic study found that *FGA* and *GSN* protein levels were upregulated in patients with OC. However, on the other hand, *FGG* and *LBP* were shown to be downregulated. It was, then, observed that high miRNA expression of *FGG* or *LBP* was associated with a shorter progression-free survival, indicating a poor prognosis for patients with epithelial cancer. Interactions between differentially expressed genes (DEGs) were noted using gene ontology analysis, the Kyoto Encyclopedia of Genes and Genomes (KEGG) pathway, and the protein–protein interaction network. This showed that hypercoagulation in epithelial OC is linked to the DEGs. The result of this study was the identification of four diagnostic and two potential prognostic biomarkers.

It should be noted that more research should be done to validate some candidate prognostic markers in a separate cohort of patients and to confirm the functional roles of exosomes in cancer.

## 6. Proteomics in the Treatment of Ovarian Cancer

Whilst most proteomic research focuses on the diagnosis and prognostic indicators for ovarian cancers, the discovery of these new biomarkers presents an exciting opportunity in the treatment and management of OC.

### 6.1. Tackling Chemotherapy Resistance

Within the UK, the standard treatment for patients with OC involves debulking surgery followed by chemotherapy, namely carboplatin and paclitaxel. However, tumour resistance to platinum-based chemotherapy presents an emerging challenge to current regimes [118]. Several groups have identified ways in which resistance occurs, many of which are related to key oncogenic signaling pathways [119].

For example, disruption of the normal functioning of the PI3K/Akt/mTOR and MAPK pathway has been linked to resistance to platinum-based chemotherapy [118,120]. Furthermore, lower expression of the presumed tumour-suppression gene RBPMS causes resistance to chemotherapy.

Whilst the over or under expression of certain proteins may indicate reduced sensitivity to chemotherapy, emerging evidence shows that targeted treatment against the pathways conferring resistance may help to overcome it [121].

For example, Lee et al. reported the involvement of CXCR4 in promoting cell dormancy in response to chemotherapy, but treatment with CXCR4 antagonists in combination with other chemotherapy agents was found to have therapeutic effects in promoting cancer cell death. Zhang et al. identified several SENP1 inhibitors that inactivate the SENP1/JAK2/STAT pathway to overcome chemotherapy resistance [122].

Table 4 summarizes biomarkers for drug resistance in OC, discovered by proteomic techniques.

### 6.2. Targeted Therapy Using Proteomics

The Cancer Genome Atlas and the International Cancer Genome Consortium have sequenced thousands of ovarian tumour specimens, which has resulted in the identification of novel genomic sequences that could be targets for therapeutic interventions [123]. Whilst much work is underway to translate this into clinical practice, the two most evidenced therapeutic agents, which are FDA license approved, are poly (ADP-ribose) polymerase (PARP) inhibitors and VEGF/VEGF receptor inhibitors [124].

## 7. PARP Inhibitors

PARP inhibitors are enzymes involved in DNA repair and require several other homologous recombination proteins for its function, notably *BRCA1* and *BRCA2*. In OC involving *BRCA* mutations, inhibition of PARP causes chromosomal instability [125], due to the activation of other error-prone DNA-repair pathways and, ultimately, leads to cell death [126]. Luo and Keyomarsi present the four main PARP inhibitors that are currently approved for use [127]. Unfortunately, however, there is evidence that many patients who are treated with PARP-inhibitors confer resistance to the treatment, presenting a challenge to its clinical use [128].

## 8. VEGF/VEGF Receptor Inhibitors

VEGF and its receptor (VEGFR) are critical in the role of angiogenesis and, hence, tumour survival. Bevacizumab, a monoclonal antibody that is a potent inhibitor of this process, has been demonstrated to improve progression-free survival and quality of life in patients with OC, and is recommended as an adjunct treatment following standard chemotherapy [129].

Whilst it is unlikely that targeted proteomic treatments will be applied as monotherapy, there is growing evidence that, if used as an adjuvant to standard protocol, they may enhance the effectiveness of treatment. This is certainly encouraging in tackling the heterogeneity of OC. Though the most promising therapies involving proteomics in OC are in early-phase clinical trials, more studies are required in the future to determine their effectiveness in the clinical setting.

## 9. Conclusions

Proteomics yields useful information regarding the identity, expression levels, modification, and interaction of proteins in the pathophysiological environment, thereby consolidating its place in the molecular sciences. Cancer involves aberrant cellular proliferation secondary to dysregulation of the cell cycle, due to a harbinger of genetic alterations. Proteomics can identify protein targets and signalling pathways related to the growth and metastasis of cancer cells. With the advent of MS-based protein analysis technology, high throughput proteomic characterisation of biological specimens is now possible. This has led to the advent of easily accessible global cancer proteome databases, via integration with bioinformatics. However, in OC—especially the high-grade serous variant—a significant degree of intra-tumoural and intra-lesion heterogeneity prevails at the genomic level. Although single-cell spatially oriented proteomics are underway, sensitivity and specificity still seem to be limiting factors in attempts to capture heterogeneity. For a cancer type infamous for late diagnosis, there is an unmet need for novel biomarkers to enable early disease detection. The evolution of quantitative proteomics is certainly bridging the gap between biomarker discovery and validation. At the same time, intricate cellular mechanisms triggering oncogenesis are a product of proteomic, transcriptomic, genomic, and epigenetic changes. Hence, integrated omics in the future can serve as the basis of the development of novel multi-omics clinical diagnostics, ensuring successful translation into clinical use and transition to precision medicine.

## Figures and Tables

**Figure 1 proteomes-10-00016-f001:**
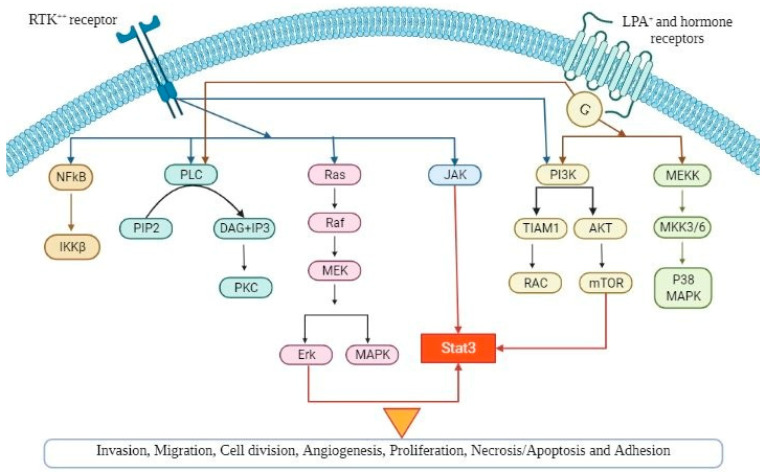
A schematic diagram demonstrating the various signaling pathways implicated in ovarian cancer pathogenesis, which, if dysregulated, can lead to tumour progression (angiogenesis, cellular hyperproliferation, resistance to apoptosis). Abbreviations are explained in the text.

**Figure 2 proteomes-10-00016-f002:**
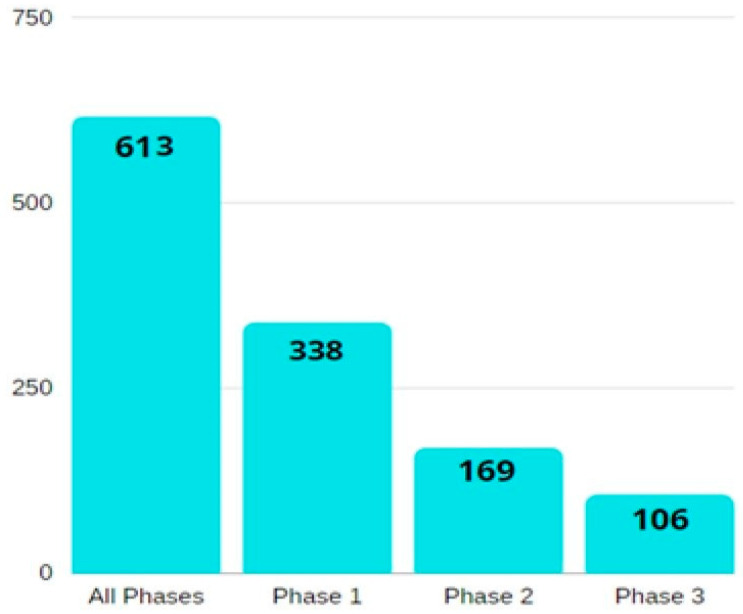
A bar diagram showing the number of phase-wise studies on ovarian cancer biomarkers. All the information is obtained from the National Cancer Institute’s EDRN website (https://edrn.nci.nih.gov/data-and-resources/biomarkers), accessed on 26 February 2022.

**Table 1 proteomes-10-00016-t001:** Proteomic biomarkers with their respective source and discovery platforms.

Approval Status	Biomarker	Sample	Sensitivity	Specificity	References	Discovery
FDA approved biomarkers	CA125	Serum/Plasma	60–70%	94%	[15,61,62,63]	Immunoassay-1981
	HE4	Serum/Plasma/Urine	72.9%	94%	[61,64,65,66]	ctDNA arrays, Immunoassay-2008
	CancerSEEK	Blood	98%	99%	[63,67]	ctDNA, Multiplex PCR assays, others
	ROMA	Serum	79%	78%	[15,63,68]	Immunoassays, Menopausal status-2010
	OVA1 (Transthyretin, β-2 microglobulin, CA125, transferrin and apolipoprotein A1)	Blood	94%	54%	[61,69]	Multivariate Immunoassay-2009
	OVERA (HE4, FSH, CA125, transferrin and apolipoprotein A1)	Blood	91–94%	69%	[15,61,63]	Multivariate Immunoassay-2016
Other biomarker candidates	Anti-TP53, TRIM-21, NY-ESO-1 (CTAG-1A) and PAX-8	Serum	46–67%	94–98%	[63,70]	PCR, line BLOT, ctDNA, Western blot, ELISA
	HE4 antigen-autoantibody complexes with CA125	Serum	38% alone 60–80% when combined	98%	[63,71]	Multiplexed Immunoassay
	MiRNAs (multiple)	Tumour/Serum/Plasma	Negative predictive value 78.6%	Positive predictive value 91.3%	[63,72,73]	Microarrays, PCR
	Kallikrein	Serum	21–26%	94%	[74]	PCR, Densitometry, DNA sequencing
	APC, RASSF1A, CHDH1, RUNX3, TFP12, SRP5 and OPCML	Serum	85%	91%	[63,75]	DNA methylation, PCR
	CA125, osteopontin, macrophage inhibitory factor and anti-IL8 autoantibodies	Serum	82%	98%	[63,76]	Multiplexed immunoassay
	CA125, apolipoprotein B, transthyretin	Serum	74%	97%	[77]	SELDI TOF, immunoassay

**Table 2 proteomes-10-00016-t002:** Ovarian cancer biomarkers as part of prospective cohort studies.

Biomarker	Discovery	Sample	No. of Patients	No. of Controls	Sensitivity	Specificity	References
CA125, C-Reactive protein, Serum amyloid A, IL-6, IL-8	Multiplexed assay	Plasma	150	212	94%	91%	[78]
Four lipid metabolites	LC-MS	Plasma	50	50	95%	35%	[79]
CA125, HE4, CA72.4, and CA15.3 (European EPIC cohort)	Immunoassay	Blood	810	1939	95%	98%	[80]
c17orf64, IRX2, TUBB6	Genome-wide methylation analysis, qMSP assays	Tissue	23	36	100%	100%	[81]
92 proteins (CA125, HE4, FOLR1, KLK11, WISP1, MDK, CXCL13, MSLN, ADAM8 were significant)	Multi assay	Blood	91	180	AUC > 0.70 for the 9 proteins	Not reported	[82]
metabolites	UPLC-MS	Plasma	140	308	Not reported	Not reported	[83]
TRIM21, NY-ESO-1, TP53, PAX8	ELISA, Western Blot	Serum	114	50	46–67%	94–98%	[70]
miR-1246, miR-595, miR-2278	Microarray, RT-qPCR	Serum/Tissue	168	65	87%	77%	[84]
10-miRNA profile (miR-320a, miR-665, miR-3184-5p, miR-6717-5p, miR-4459, miR-6076, miR-3195, miR-1275, miR-3185, miR-4640-5p)	Microarray	Serum	428	2759	99%	100%	[85]
HE4 autoantibody	Immunoassay	Serum	145	212	38% alone, 60–80% when combined	98%	[71]
lncRNAs	Microarray, qPCR	Tissue	18	31	Not reported	Not reported	[86]

**Table 3 proteomes-10-00016-t003:** Mechanistic biomarkers in ovarian cancer, with reference to patient sample-based studies.

Biomarkers		References
Protein antigen	CA125, HE4, CA72.4, CA15-3, CEA and V-CAM1 Glycodelin, E-cadherin and IL-639 or transthyretin	[63]
Immune related-Cytokine, chemokine	IL-6, IL-7, IL-8, IL-12, B7-H3, B7-H4 interferon-γ, auto antibodies against TP53, TRIM-21, NY-ESO-1 (CTAG-1A), PAX-8	[63,87]
Signalling molecule	EGFR, HER2, p53 mutaion, cyclin D1, cyclin E, sFas	[87]
Inherited gene mutations	BRCA1, BRCA2 MSH2, MLH1, MSH6, PMS2 RAD51C, RAD51D, BRIP1, BARD1, CHEK2, MPE11A, NBN, PALB2, RAD50, TP53	[88]
Gene expression	CA125, osteopontin, kallikrein 10, secretory leukoprostease inhibitor, matrix metalloproteinase-7 FOL3, survivin, MCM3, E2Fs, VTCN1, SYNE1, AKAP14, KNDC1, DLEC1 ovarian cancer prognostic profile (115 gene signature) ctDNA: APC, RASSF1A, CHDH1, RUNX3, TFP12, SRP5, OPCML	[89,90]
Angiogenesis	VEGF, FGF-1, Claudin-3, claudin-7, EZH2, EphA2	[87]
Epigenetic changes	Hypermethylation: BRCA1, RASSF1A, APC, p14ARF, p16INK4a, DAPKinase59 ARMCH1, ICAM4, LOC134466, PEG3, PYCARD SGNE160 MiRNAs: miR-200a, miR-141, miR-199a, miR-140, miR-145, miR-125b163 miR-18266, miR-21, miR-92, miR-93, miR-126, miR-29a, miR-155, miR-127, miR-99b68	[88]
Protein antigen	CA125, HE4, CA72.4, CA15-3, CEA and V-CAM1 Glycodelin, E-cadherin, IL-639, or transthyretin.	[63]

**Table 4 proteomes-10-00016-t004:** Drug resistance markers.

Biomarkers	Techniques	Reference
Annexin3, Destin	MALDI-TOF	[44]
ERp57	MALDI-TOF, ESI-Q-TOF	[44]
Activated Leucocyte Cell Adhesion Molecule, Nestin	Orbitrap	[44]
Pyruvate kinase isozymes M1/M2 Heat shock protein family D	ESI-Q-TOF	[44]
Abbreviation: ESI-Q-TOF; Electrospray-ionisation quadrupole time-of-flight mass spectrometry		

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
