# Peer review of "Applications of Proteomics in Ovarian Cancer: Dawn of a New Era"

_proteomes, 2022, doi:10.3390/proteomes10020016_

Round 1
Reviewer 1 Report
A brief description of the caption of FIG. 1 is needed. At present, there’s the only title of the figure legend.
Author Response
Dear Editor and Reviewers,
I am pleased to resubmit for publication the revised version of manuscript entitled “Applications of Proteomics in Ovarian Cancer: Dawn of a New Era”.
Thankfully the reviewers provided us with a great deal of guidance, regarding how to better position the article. We are hopeful you agree that this revision will update our comprehensive review. All the comments have been addressed, as shown in the revised version of the manuscript, along with this point-by-point response to the reviewers' comments.
All corresponding are blue changes in the manuscript.
Reviewer #1:
A brief description of the caption of FIG. 1 is needed. At present, there’s the only title of the figure legend.
Response:
Thank you for your comment. We have put a brief description onto Fig.1 – Lines 98-100. Abbreviations as per main text.
Reviewer 2 Report
Ghose et.al. have presented and worked on an important topic. However, there are following major concerns needed to be addressed:
- Please provide a detailed table summarizing the biomarker (from fluid-serum, plasma) discovery using proteomics (different platforms) in different population.
- Please highlight whether these biomarkers have been validated in a prospective cohort or not?
- State the pros and limitation of the studies performed till date. Discuss the alternate approach in design.
- Highlight in a tabular format regarding the mechanistic discovery phase studies in ovarian cancer (use patient sample based and animal model based studies separately).
- Please discuss how targeted proteomics can be more relevant in clinical diagnosis setting.
Author Response
Dear Editor and Reviewers,
I am pleased to resubmit for publication the revised version of manuscript entitled “Applications of Proteomics in Ovarian Cancer: Dawn of a New Era”.
Thankfully the reviewers provided us with a great deal of guidance, regarding how to better position the article. We are hopeful you agree that this revision will update our comprehensive review. All the comments have been addressed, as shown in the revised version of the manuscript, along with this point-by-point response to the reviewers' comments.
All corresponding are blue changes in the manuscript.
Reviewer #2:
Ghose et al have presented and worked on an important topic. However, there are following major concerns needed to be addressed:
-
Please provide a detailed table summarizing the biomarker (from fluid-serum, plasma) discovery using proteomics (different platforms) in different population.
-
Please highlight whether these biomarkers have been validated in a prospective cohort or not?
-
State the pros and limitation of the studies performed till date. Discuss the alternate approach in design.
-
Highlight in a tabular format regarding the mechanistic discovery phase studies in ovarian cancer (use patient sample based and animal model based studies separately).
-
Please discuss how targeted proteomics can be more relevant in clinical diagnosis setting.
Response:
Thank you for your suggestions.
Point 1; Refer to Table 1
Point 2 – 3; Refer to Table 2
Point 4; Refer to Table 3
We have discussed Point 5 – relevance of targeted proteomics in clinical diagnosis setting – Lines 607-620.
Reviewer 3 Report
In this review article, the authors give a general introduction to the biomarker discovery of OC based on proteomics. In general, the content is sufficient. The authors spent quite a large volume of text describing the methodology, such as LC-MS and MALDI-TOF. Based on the title, I would expect the authors to talk about how the discovery of various biomarkers will guide the treatment and management of OC or uncover novel pathogenic mechanisms of OC. But the information is absent.
It is okay for the authors to mention the historical story of OC’s biomarker discovery, particularly those identified from proteomics studies. However, the authors may need to revise the title.
In figure 2, the authors showed the number of biomarkers of OC in each of the phases. The authors may consider using a table to indicate which biomarkers have been discovered based on proteomics. The authors do not need to show all of them but select some.
Also, the authors may need to provide the sensitivity and specificity of the biomarkers, e.g. CD81, TSG101, miRNAs, etc. The information will show the reliability of the biomarkers.
Author Response
Dear Editor and Reviewers,
I am pleased to resubmit for publication the revised version of manuscript entitled “Applications of Proteomics in Ovarian Cancer: Dawn of a New Era”.
Thankfully the reviewers provided us with a great deal of guidance, regarding how to better position the article. We are hopeful you agree that this revision will update our comprehensive review. All the comments have been addressed, as shown in the revised version of the manuscript, along with this point-by-point response to the reviewers' comments.
All corresponding are blue changes in the manuscript.
Reviewer #3:
In this review article, the authors give a general introduction to the biomarker discovery of OC based on proteomics. In general, the content is sufficient. The authors spent quite a large volume of text describing the methodology, such as LC-MS and MALDI-TOF. Based on the title, I would expect the authors to talk about how the discovery of various biomarkers will guide the treatment and management of OC or uncover novel pathogenic mechanisms of OC. But the information is absent.
It is okay for the authors to mention the historical story of OC’s biomarker discovery, particularly those identified from proteomics studies. However, the authors may need to revise the title.
In figure 2, the authors showed the number of biomarkers of OC in each of the phases. The authors may consider using a table to indicate which biomarkers have been discovered based on proteomics. The authors do not need to show all of them but select some.
Also, the authors may need to provide the sensitivity and specificity of the biomarkers, e.g. CD81, TSG101, miRNAs, etc. The information will show the reliability of the biomarkers.
Response:
Thank you for your suggestions.
We have added a section entitled "Proteomics in the treatment of ovarian cancer" – Lines 680 - 736. Moreover, table 4 is linked to that.
Also as per reviewer, tables with respect to proteomic biomarkers and sensitivity and specificity can be found in Table 1 and 2.
If the reviewer feels that the change of the title is not a major concern, we would prefer to maintain the current title.
Round 2
Reviewer 2 Report
Accepted
Reviewer 3 Report
The authors have already addressed all my concerns.